# Inclusion in Uncertain Times: Changes in Practices, Perceptions, and Attitudes around Accessibility and Inclusive Practice in Higher Education

**Kate Lister [1],\*** , **Victoria K. Pearson [2]** , **Tim Coughlan [3] and Felipe Tessarolo [3]**

1   Faculty of Learning and Teaching, Arden University, Coventry CV3 4FJ, UK
2   Faculty of STEM, The Open University, Milton Keynes MK7 6AA, UK
3   Institute of Educational Technology, The Open University, Milton Keynes MK7 6AA, UK
\*   Correspondence: klister@arden.ac.uk

**Abstract:** Accessibility, inclusive teaching, and student support are multi-faceted; they are dependent on wider institutional factors, such as leadership, resource, systems, and culture. To be truly inclusive requires a whole institution approach, with voices, perspectives, and stakeholder buy-in sought from across the institution. This can be extremely challenging because these can be sensitive to myriad institutional, sector, and societal changes that can influence working practices, resource management, and capacity. In this paper, we analyse responses from three iterations of a biennial staff survey conducted in 2017, 2019, and 2021 at the Open University (OU), capturing views on accessibility and inclusion before and during the COVID-19 pandemic and other challenging circumstances. These responses, from tutors, module and programme teams, educational technologists, library staff, and student support teams, reveal crucial insight into the (in)accessibility of support and practice across the institution, as well as insight into staff skills, attitudes, and knowledge around accessibility, and the fitness for purpose of the systems and structures in place. In this analysis, we explore how staff practices and perceptions change over time; identify the themes that remain constant over time, despite global circumstances; and explore how these themes can inform a whole-institution approach to accessibility and inclusion.

**Keywords:** accessibility; inclusion; education; students; survey; change; COVID-19

## 1. Introduction

For over 30 years, higher education institutions have been steering a path towards greater accessibility and inclusion in higher education, and greater equity for students with disabilities. It is fair to say the journey has not always been smooth sailing; UK institutions have been buffeted by winds of changing legislation, policy changes, and a global pandemic, and there is still a participation gap, with only 14.3% of the UK HE student body disclosing a disability [1], compared with 19% in the wider UK population [2]. In this paper, we analyse survey responses from three staff surveys, conducted in 2017, 2019, and 2021, to chart the progress, the trials and tribulations, and the beacons of hope in this uncertain journey.

First, it is important to set out and justify the terminology we use in this paper. The language used to describe disability has long been contentious, with rigorous debates over terminology and word order [3–5]. The UK social model of disability [6] generally advocates for the words 'disability' and 'disabled' to be used, and for 'disabled' to be used attributively (before the noun, i.e., 'disabled person') rather than predicatively (after the noun, i.e., 'person with a disability'), as this is intended to show that people are disabled by inaccessible societies and circumstances rather than by their bodies. This argument, sometimes called the 'identity first' model, is robustly countered by the 'person first' model of disability, used throughout the USA and other parts of the world. This posits that people

should always be spoken about first in order to reduce bias, and the word 'disability' should be used predicatively ('person with a disability') in order that speakers see the person first, before the disability [3]. To further complicate the language issue, people with disabilities are (of course) not a single homogenous group with the same feelings and beliefs; language around disability is strongly linked to identity, and each individual has their own preferences. When exploring language preferences in disabled students, Lister et al. found that overall there was not a strong inclination towards a single model of language, that different people had different preferences, and that these depended on contexts [5]. Therefore, in this paper, we use 'disabled students' and 'students with disabilities' interchangeably.

It is also important to define 'disability' in a UK higher education context. The UK Equality Act defines disability as 'a physical or mental impairment that has a 'substantial' and 'long-term' negative effect on your ability to do normal daily activities' [7]. In UK higher education, disability is broken down into a number of categories; the wording and precise categories can change slightly between different institutions, but the generally accepted categories are as follows:

- Specific learning difficulties, such as dyslexia.
- Social or communication impairments, such as autistic spectrum conditions.
- Unseen or long-standing illness or health conditions, such as cancer, diabetes or epilepsy.
- Mental health conditions.
- Physical impairment or mobility issues.
- Deaf or serious hearing impairment.
- Blind or serious visual impairment uncorrected by glasses.
- Speech or language impairment.
- Other: a disability, impairment or medical condition that is not listed [8].

There is increasing recognition of public sector duty towards disabled people, and the last decade has seen huge progress in support of students with disabilities. The passing of the Equalities Act (2010), which supported equitable participation in higher education, led to significant work in universities to embed inclusive approaches to education and identify reasonable adjustments. In addition, the Disabled Student Allowance, the grant awarded to students with declared disabilities, was reformed and limited, excluding any costs that may be considered the responsibility of higher education institutions (HEIs), e.g., adjustments to teaching and learning including printing, non-medical helpers. The increased marketisation of UK higher education (HE) also provoked changes for universities to be inclusive for all, with consumer-focused approaches and market-driven incentives. However, with this increasingly competitive market, HEIs have been required to report on key metrics that enable consumers to make choices and to drive funding allocations (e.g., frameworks for excellence such as TEF, REF and KEF). This has included a requirement for HEIs to develop Access and Participation Plans that outline provisions to support the progress of underrepresented students through and beyond their higher education experiences.

Within this environment of change and within a wider climate of austerity and budget cuts, the introduction of new legislation on accessibility for public bodies as part of an EU directive [9] has been both extremely welcome and fundamentally challenging for many institutions. This legislation aims 'to ensure that all citizens can access services and participate in society, and promote and facilitate accessible digital developments; and second, to mitigate the need for individuals to take legal action to ensure basic access.' [10]. In practice, this means institutions are required to take steps towards meeting the Web Content Accessibility Guidelines (WCAG) 2.1 level AA as the recognised standards for their web-based content, to provide public accessibility statements about how accessible their content is, and provide a feedback mechanism for people who find something inaccessible [9]. Effectively, it means 'the success or failure of digital accessibility in the public sector will be centrally monitored for the first time' [10]. This was greeted by accessibility practitioners in higher education both with elation and trepidation; it is undoubtedly welcome legislation and will result in much more inclusive higher education environments, but in many cases the level

of work required to meet the standards was prohibitive. Furthermore, web-based content is only a small part of teaching and learning, and the legislation does not cover areas such as pedagogy, assessment design, tuition, and other non-digital aspects of teaching.

In our own institution, The Open University (OU), UK, there was an additional burden of upheaval on staff in this time. Between 2015 and 2018, the OU leadership team expounded upon the sector discourse of austerity with an institutional strategy of redundancies, curriculum review, cuts, and closures. While first met with 'quiescence' from staff, with related emotions of 'fear, anger, cynicism, despair and possibly depression' [11], the response from some staff slowly turned to anger, then opposition, bi-directional hostility, and finally to organised resistance. This culminated in the resignation of the vice-chancellor in 2018, a review of strategy and leadership team under a temporary vice-chancellor between 2018 and 2019, and finally a stable leadership team from 2019 onward. The impact of these events on many university staff is not to be overlooked. Bowes Catton et al. talk about the 'profound fear' many staff felt during this time, describing it eloquently as 'an erosion of our soul' [11]. These events dominated the working lives, thoughts, and energy of many university staff for a long period of time, and it is difficult to encourage an agenda of inclusion and accessibility in these circumstances.

Following the upheaval caused by institutional changes and the new accessibility legislation was a rather larger-scale societal change: the COVID-19 pandemic. Even in a distance-learning institution, this created mass consternation and an enormous amount of work, as all the face-to-face working and assessment practices had to be rapidly migrated online. The impacts of COVID-19 on students and academics are well documented [12], [13], but there were also substantial impacts on academic-related and professional services staff, who also have responsibility for accessibility and inclusion. It is important that views and responses from a wide range of staff are captured; studies seeking to measure accessibility and inclusion tend to focus on faculty voices (e.g., [14–16]), or student voices (e.g., [17]), and miss the crucial insight that professional services staff such as learning technologists, librarians, and student support staff can add, and the challenges they experience in their own contexts.

The cumulative impact of all these institutional, sector, and societal challenges meant challenges for furthering the agenda of accessibility and inclusion. Many university staff were stressed, overworked, and had other priorities; in many cases, they were also emotionally exhausted and had little energy to devote to passion projects or goodwill causes (which is often how accessibility is positioned). However, during this time, various interventions, additional training, and awareness-raising initiatives took place. These included a university-wide 'mental wellbeing in distance learning' project, which included focus groups and pilot projects [18–20]; a university mental health strategy; a collaborative project to co-create guidance around designing learning for autistic and neurodiverse learners [21]; a project to co-create guidance around inclusive language regarding disability [22]; and an 'accessibility champions' initiative, accompanied by training and written guidance, to support learning technologists with guidance and human support in relation to accessibility and inclusion [23,24]. Studies have found that targeted training programmes for staff can have a positive impact on attitudes and practices [25,26], so it is reasonable to assume that, despite the challenging circumstances, these initiatives may still have had an impact in raising awareness or furthering activity in accessibility and inclusion. This paper investigates this, using the results of three biennial staff surveys spanning 2017–2021 to identify how staff knowledge, skills, attitudes, and support contexts with regard to accessibility and inclusive practice were affected in the Open University during this challenging time.

*Institutional Context*

The Open University (OU) has over 150,000 undergraduate and taught postgraduate students studying part time and at a distance, and at the time of writing, over 33,000 students (21.9% of the total cohort) had disclosed disabilities to the institution. Most students are mature and have a variety of other life commitments [27]. Additionally, as the

University operates an open-entry policy that does not require students to have any pre-requisite academic qualifications, students may have little or no previous experience of post-secondary education.

OU courses are delivered fully or partly online and are sometimes accompanied by printed materials. Students work through curriculum content, library materials, and asynchronous activities independently, supported by tutors (also called associate lecturers, or ALs) within a tutor group. Tutors run tutorials for their groups, provide academic support to students, and mark and provide feedback on assessment. The production of courses is a collaborative effort, where academics work with curriculum managers, media developers, digital media specialists, librarians, and experts in technology-enhanced learning, and may spend a year or more producing a course. As such, learning and teaching is very much a team activity, with multiple stakeholders involved, all of whom have a role to play in accessibility and inclusion (in line with the model of accessible e-learning practice proposed by [28]). This survey aims to better understand the views of some of these different stakeholder groups.

## 2. Methodology

Every 2 years, between 2017 and 2021, a survey was sent to a cross-sectional sample of university staff. This aimed to measure their perceptions of their knowledge, skills, attitudes, and support context, in terms of accessibility and inclusive practice in their roles, and how these changed over time. 'Skills, knowledge and attitudes' are commonly measured indicators of job proficiency [29], and as inclusive practice requires "integrated approaches to inclusion, which consider the roles of all members of campus communities in working towards this goal" [30], staff perceptions of their support context were also measured in the surveys, in terms of the training, guidance, and human support they receive, as well as the sense they have of the institutional commitment to accessibility and inclusive practice.

The results of the first survey are shared in [24]. As explained there, the survey was designed for this context by a team of academics; the team did not use an existing scale or measure as there was not one available that would meet the needs of this study. The instrument design took place iteratively over a period of several weeks, with input from stakeholders both within the project team and more broadly within the institution, and revisions made to drafts accordingly. Ethical approval was granted by the OU's Human Research and Ethics Committee, and the method and survey instrument were reviewed in detail and approved by the OU's Staff Survey Project Panel. This process ensured a robust check on the approach, methodology, survey instrument, language, and sample. As explained in [24], an initial pilot was carried out with a sample of 42 staff before the first iteration of the survey. These measures ensured the validity and reliability of the survey for the OU context, but the authors recommend that any parties wishing to replicate this study should adapt the survey instrument for their own context and test reliability and validity accordingly.

The survey consisted of four sections.

- Part 1: demographic information (including unit, role, and longevity of service) (three items).
- Part 2: general statements about accessibility and inclusive practice with five-point Likert scale response options, plus 'not relevant' (26 items).
- Part 3: role specific statements, unique to different staff groups, with five-point Likert scale response options (five to eight items).
- Part 4: a matrix on levels of confidence in supporting students with different disability types (one item with sub-parts).

The surveys were distributed at the following times (due to institutional circumstances, this could not be within the same timeframe in each iteration).

- July–November 2017.

- August–September 2019.
- June–July 2021.

*Participants*

Due to the particular context, size, and teaching model at the Open University, five different groups of staff were surveyed:

1.  Academics and curriculum managers (i.e., staff who design and create module content and therefore have responsibility for inclusive and accessibility pedagogy and curriculum design).
2.  Learning technologists (i.e., staff who advise and work with module teams, who create and deliver digital media, and who manage the virtual learning environment; these staff often make decisions and recommendations regarding digital accessibility and inclusive learning design).
3.  Student support teams (including the disability support team) who work on the front line managing reasonable adjustment requests and queries or problems relating to accessibility.
4.  Associate lecturers (also called tutors, these staff teach, run tutorials and learning events, provide academic support to students, and mark their assessments).
5.  Library staff (who provide books and articles for modules, arrange alternative formats of library content, and support students with accessibility issues of books and journal articles).

Samples of the staff groups were selected by the OU's People Services team and were invited to survey by email. Due to institutional staff attrition, and the fact that only samples of staff were surveyed rather than all staff in an area, it is unlikely that responses received are from the same individual staff across the period. All groups were given 4 weeks to respond, with a reminder sent 2 weeks into the survey period. There were two exceptions to this; in the first iteration of the survey, library staff were not surveyed (they opted in in the second iteration), and tutors were not invited by email but received an open invitation via their intranet home page, due to institutional issues around tutor workload and volume of email requests at that time [24]. From the second iteration of the survey onwards, samples of tutors and library staff were invited, in line with the other groups.

The numbers invited to participate, the responses received, and the calculated response rate are shown in Table 1.

**Table 1.** Participant response rates.

|      | Participants | Academics | Learning Technologists | Student Support | Associate Lecturers | Library |
|------|--------------|-----------|------------------------|-----------------|---------------------|---------|
|      | Sent to | 871 | 248 | 251 | (Open invitation) | - |
| 2017 | Responses | 261 | 57 | 82 | 66 | NA |
|      | Response rate | 29.97% | 23% | 32.67% | NA | NA |
|      | Sent to | 895 | 209 | 300 | 900 | 66 |
| 2019 | Responses | 255 | 73 | 54 | 292 | 38 |
|      | Response rate | 28% | 35% | 18% | 32% | 58% |
|      | Sent to | 800 | 200 | 300 | 800 | 70 |
| 2021 | Responses | 196 | 54 | 116 | 274 | 26 |
|      | Response rate | 24.50% | 27% | 38.67% | 34.25% | 37.14% |

Survey results were analysed in SPSS using inferential statistics to test independence of responses between groups. Pearson's chi-squared was used to determine statistical significance, with an alpha level of 0.05 for all statistical tests. Responses of 'not applicable' were considered a non-response and were discarded.

### 3. Findings

The findings from the surveys are categorised as relating to staff knowledge, attitudes, skills, and support contexts, and are presented below.

#### 3.1. Knowledge

Staff were asked about their knowledge of accessibility and inclusive practice in two ways: they were asked about their perceptions of their levels of knowledge, and their knowledge was tested with questions that had correct or incorrect answers.

#### 3.1.1. Perceptions of Knowledge

Staff perceptions of their levels of knowledge relating to disability groups, accessibility issues, reasonable adjustments, and basic legal rights of disabled students remained consistently high over the three survey iterations, with an average of 85.4% of staff agreeing with the statements. These questions showed only non-significant increases or decreases over time for all staff groups (shown in Table 2).

**Table 2.** Staff self-perceptions of accessibility knowledge.

| Question | Staff Group | %Agree/Strongly Agree | | |
| | | 2017 | 2019 | 2021 |
|---|---|---|---|---|
| 4. I am aware of the types of conditions the OU classifies as disabilities. | Academics | 87.4% | 88.6% | 87.8% |
| | ALs | 90.9% | 90.4% | 89.8% |
| | Learning Tech | 68.4% | 84.9% | 88.9% |
| | Library | | 92.1% | 96.2% |
| | Student support | 98.8% | 94.4% | 95.7% |
| 5. I am aware of the type of accessibility issues disabled students can face. | Academics | 93.1% | 91.8% | 90.3% |
| | ALs | 92.4% | 94.2% | 91.6% |
| | Learning Tech | 87.7% | 97.3% | 94.4% |
| | Library | | 97.4% | 100.0% |
| | Student support | 98.8% | 94.4% | 88.8% |
| 6. I am aware of the basic legal rights disabled students have in a university context. | Academics | 74.7% | 74.9% | 77.6% |
| | ALs | 71.2% | 73.3% | 75.9% |
| | Learning Tech | 59.6% | 74.0% | 75.9% |
| | Library | | 73.7% | 96.2% |
| | Student support | 76.8% | 70.4% | 81.0% |
| 26. I know what is meant by the term 'reasonable adjustments'. | Academics | 83.9% | 83.5% | 85.7% |
| | ALs | 83.3% | 79.5% | 77.7% |
| | Learning Tech | 71.9% | 82.2% | 68.5% |
| | Library | | 81.6% | 76.9% |
| | Student support | 95.1% | 94.4% | 94.0% |

The lower scoring questions, relating to staff perceptions of their knowledge of the role of the Disability Support Team, and about where to find guidance on inclusive practice, both showed significant increases in knowledge over time and significant variations between staff groups. In both questions, increases were driven by specific staff groups.

For question 11, 'I know where to find information and guidance about inclusive and accessible practice', learning technologists showed a significant increase in knowledge, rising by 23.1 percentage points over the 5 years ($X^2$ (4, $N$ = 184) =11.807, $p$ = 0.019). This followed a targeted intervention in 2018 providing written guidance, training, and human support ('accessibility champions'), and is likely to be in response to that. Student support teams showed a sharp increase of 22.3 percentage points between 2017 and 2019, but little change in 2021 ($X^2$ (4, $N$ = 252) = 16.410, $p$ = 0.003). All other groups showed only non-significant increases or decreases in this time.

For question 19, 'I am aware of the nature and extent of the role undertaken by the OU's Disability Support Team', it was ALs driving the change, showing an increase of

19 percentage points over the 5 years ($X^2$ (4, $N$ = 632) = 15.528, $p$ = 0.004). This is likely due to increased communication from the Disability Support Team, with a focus on ALs as an audience. Figures for questions 11 and 19 are shown in Table 3.

**Table 3.** Staff responses to questions of guidance and support.

| Question | Staff Group | %Agree/Strongly Agree | | |
|---|---|---|---|---|
| | | **2017** | **2019** | **2021** |
| 11. I know where to find information and guidance about inclusive and accessible practice. | Academics | 61.3% | 62.4% | 63.3% |
| | ALs | 69.7% | 68.8% | 65.7% |
| | Learning Tech | 49.1% | 58.9% | 72.2% |
| | Library | | 78.9% | 92.3% |
| | Student support | 61.0% | 83.3% | 81.0% |
| 19. I am aware of the nature and extent of the role undertaken by the OU's Disability Support Team. | Academics | 52.9% | 53.3% | 54.1% |
| | ALs | 39.4% | 46.6% | 58.4% |
| | Learning Tech | 38.6% | 41.1% | 38.9% |
| | Library | | 52.6% | 73.1% |
| | Student support | 68.3% | 64.8% | 70.7% |

### 3.1.2. Tested Knowledge

Staff knowledge of attainment (Q. 27) and employment gaps (Q. 28) were broadly static between 2017 and 2019, and then rose significantly in 2021 for all staff groups ($X^2$ (4, $N$ = 1844) = 125.067, $p$ < 0.001), except for Library staff, who still rose but non-significantly ($X^2$ (2, $N$ = 64) = 4.996, $p$ = 0.082) These were low-scoring questions in 2017 and 2019; on average, 27.5% of staff gave a correct answer in 2017, 22.9% gave a correct answer in 2019, but 37.2% gave a correct answer in 2021.

While there were differences between staff groups, for example with the learning technologists showing the largest rise in awareness (24.3 percentage points on Q. 27) and ALs showing lower increases (around 6 percentage points on both questions), this was overall an institution-wide increase in knowledge of inequalities. It may have been the impact of the new leadership team and subsequent increased discourse around inequalities, or could have been a result of the pandemic and increased sector discourse and awareness. In contrast to this, staff awareness of accessibility challenges caused by practical work (Q. 25) did not show significant changes over time. Data is shown in Table 4.

**Table 4.** Staff responses to questions testing accessibility knowledge.

| Question | Staff Group | %Correct Answer Given | | |
|---|---|---|---|---|
| | | **2017** | **2019** | **2021** |
| 27. Students with disabilities are just as likely to gain a good degree (2i or higher) as students without disabilities. | Academics | 22.6% | 23.5% | 40.8% |
| | ALs | 13.6% | 6.8% | 20.1% |
| | Learning Tech | 3.5% | 17.8% | 27.8% |
| | Library | | 10.5% | 23.1% |
| | Student support | 18.3% | 9.3% | 31.0% |
| 28. Students with disabilities are just as likely to gain professional employment as students without disabilities. | Academics | 43.7% | 40.8% | 50.5% |
| | ALs | 48.5% | 37.0% | 43.1% |
| | Learning Tech | 33.3% | 26.0% | 46.3% |
| | Library | | 26.3% | 42.3% |
| | Student support | 36.6% | 31.5% | 47.4% |
| 25. Practical work, such as fieldwork and labwork, can present challenges for students with disabilities. | Academics | 67.4% | 74.9% | 71.4% |
| | ALs | 68.2% | 66.1% | 66.1% |
| | Learning Tech | 63.2% | 74.0% | 77.8% |
| | Library | | 57.9% | 69.2% |
| | Student support | 80.5% | 79.6% | 73.3% |

Overall, where staff knowledge (either perceived and tested) was high, it remained stable, while areas where staff knowledge or perceptions of knowledge had been lower, either for particular groups or throughout the university, increases were observed. There were no significant decreases, but there were periods of stagnation where low levels of knowledge were not increased between certain iterations of the survey. The highest increases were observed in the learning technologists' responses, although ALs and student support teams also showed increases in some areas.

*3.2. Attitudes*

Staff personal commitment to inclusion remained high, with an average of 94.4% of staff agreeing or strongly agreeing that they felt personally committed to accessibility (Q. 7) and no significant increases or decreases. Similarly, staff agreement that 'all staff have a responsibility to support students with disabilities' (Q. 14) also remained high overall, but showed a significant increase of 10.9 percentage points from the ALs between 2017 and 2019 ($X^2$ (4, $N$ = 632) = 14.697, $p$ = 0.005). Staff agreement that 'all teaching and learning activities should be made inclusive and accessible to all students' (Q. 18) also remained steady for most and also showed a significant increase from the ALs between 2017 and 2019 ($X^2$ (4, $N$ = 632) = 17.572, $p$ = 0.001). Finally, staff belief that 'support we provide to disabled students at University will equip them to deal with the world of work' (Q. 29) remained consistently low; on average, only 43.4% of staff agreed with the statement, and although rises were observed (particularly in the ALs), these were not statistically significant. The substantially lower agreement rates with Q. 29 may be interpreted as a lack of faith in the commitment shown to accessibility in the wider world, beyond the university. Data is shown in Table 5.

**Table 5.** Staff attitudes towards accessibility.

| Question | Staff Group | %Agree/Strongly Agree | | |
|---|---|---|---|---|
| | | 2017 | 2019 | 2021 |
| 7. I feel committed to accessibility (in my role). | Academics | 98.5% | 96.5% | 96.4% |
| | ALs | 89.4% | 96.9% | 97.1% |
| | Learning Tech | 94.7% | 93.2% | 94.4% |
| | Library | | 92.1% | 88.5% |
| | Student support | 96.3% | 94.4% | 94.0% |
| 14. All staff should have a responsibility to support students with disabilities. | Academics | 87.7% | 91.0% | 91.8% |
| | ALs | 83.3% | 94.2% | 94.2% |
| | Learning Tech | 98.2% | 86.3% | 92.6% |
| | Library | | 97.4% | 92.3% |
| | Student support | 96.3% | 96.3% | 99.1% |
| 18. All teaching and learning activities should be made inclusive and accessible to all students. | Academics | 75.5% | 79.6% | 81.6% |
| | ALs | 77.3% | 92.5% | 90.9% |
| | Learning Tech | 82.5% | 80.8% | 87.0% |
| | Library | | 97.4% | 92.3% |
| | Student support | 93.9% | 96.3% | 95.7% |
| 29. The support we provide to disabled students at the OU will equip them to deal with the world of work. | Academics | 46.0% | 45.5% | 43.4% |
| | ALs | 28.8% | 45.5% | 43.4% |
| | Learning Tech | 42.1% | 45.2% | 33.3% |
| | Library | | 55.3% | 57.7% |
| | Student support | 37.8% | 46.3% | 37.9% |

Overall, positive attitudes to accessibility within the university environment remained high, with the ALs showing significant increases in two questions between 2017 and 2019.

*3.3. Skills*

Staff confidence in their practice-related skills (supporting and signposting students, and recognising accessibility issues) remained broadly consistent, with no significant

difference between 2017 and 2019 for most staff groups. The exception to this was the learning technologists, who showed a significant increase in skills and confidence in 2019 ($X^2$ (4, $N = 390$) = 6.4007, $p = 0.040$). This was probably in response to the previously mentioned intervention in 2018 providing support and guidance. However, learning technologists then showed a sharp decrease in 2021, which was statistically significant when all three survey iterations and all three questions were taken into account ($X^2$ (4, $N = 552$) = 9.9014, $p = 0.042$). This may be due to the insecurities relating to meeting the demands of the new accessibility legislation, or working remotely during the pandemic, or to a combination of factors. The figures are shown in Table 6.

**Table 6.** Staff confidence in accessibility.

| Question | Staff Group | %Agree/Strongly Agree | | |
|---|---|---|---|---|
| | | 2017 | 2019 | 2021 |
| 8. I feel confident supporting disabled students, as far as my role requires me to. | Academics | 72.8% | 71.4% | 71.9% |
| | ALs | 69.7% | 77.4% | 77.7% |
| | Learning Tech | 64.9% | 79.5% | 64.8% |
| | Library | | 73.7% | 69.2% |
| | Student support | 84.1% | 79.6% | 81.0% |
| 9. I am confident I could signpost students with disabilities to relevant sources of support, if necessary. | Academics | 64.0% | 69.0% | 69.9% |
| | ALs | 72.7% | 71.9% | 75.2% |
| | Learning Tech | 43.9% | 50.7% | 48.1% |
| | Library | | 60.5% | 76.9% |
| | Student support | 82.9% | 87.0% | 84.5% |
| 12. I am confident that I can recognise potential accessibility issues in the context of my role. | Academics | 80.1% | 81.2% | 81.1% |
| | ALs | 68.2% | 74.0% | 70.4% |
| | Learning Tech | 86.0% | 94.5% | 83.3% |
| | Library | | 68.4% | 92.3% |
| | Student support | 87.8% | 81.5% | 81.0% |

In contrast to this, staff confidence in supporting certain different disability categories changed significantly over time and showed significant variations between staff groups. Generally speaking, staff groups who showed higher confidence in supporting certain disability groups in 2017 (mainly ALs and student support teams) showed decreases in confidence by 2021, while staff who were less confident initially (mainly academics and learning technologists) showed modest yet significant gains in confidence. However, the decreases in confidence were generally more pronounced and resulted in an overall decrease in staff confidence for all disability types except for mental health and autism. These two categories both saw significant increases in staff confidence (mental health ($X^2$ (4, $N = 1844$) = 33.569, $p < 0.001$) and autism ($X^2$ (4, $N = 1844$) = 21.298, $p < 0.001$)), probably due to training and awareness initiatives within the university, which ran from 2018 to 2021. The results are shown in Table 7; due to the size of this table, the results are heat mapped for clarity (highest percentages are green, ranging down through yellow, orange and red for the lowest).

Staff responses from this section of the survey present a fascinating contrast to the findings relating to staff knowledge around accessibility, where previously high levels of knowledge remained stable while increases were observed where knowledge levels had been low. This raises interesting questions about the differences between perceptions of knowledge and confidence in skills, especially at a granular level (i.e., by disability type.) It also raises questions about the level of support available to staff groups who showed decreases in confidence across disability categories, i.e., student support teams and ALs.

**Table 7.** Staff confidence supporting different disabilities.

| 35. I Feel Confident Supporting People with the Following Disabilities Through My Role | | | | |
|---|---|---|---|---|
| | | **%Agree/Strongly Agree** | | |
| **Disability Goup** | **Staff Group** | **2017** | **2019** | **2021** |
| Students with autism spectrum conditions | Academics | 37.5% | 48.2% | 49.5% |
| | ALs | 54.5% | 61.3% | 60.9% |
| | Learning Tech | 19.3% | 27.4% | 40.7% |
| | Library | | 50.0% | 38.5% |
| | Student support | 59.8% | 77.8% | 54.3% |
| Students with conditions causing fatigue and/or pain | Academics | 59.8% | 63.5% | 68.9% |
| | ALs | 90.9% | 85.3% | 79.6% |
| | Learning Tech | 33.3% | 28.8% | 38.9% |
| | Library | | 52.6% | 53.8% |
| | Student support | 82.9% | 81.5% | 64.7% |
| Students with hearing impairment or d/Deaf | Academics | 62.8% | 68.2% | 64.8% |
| | ALs | 77.3% | 67.1% | 64.2% |
| | Learning Tech | 75.4% | 64.4% | 72.2% |
| | Library | | 55.3% | 65.4% |
| | Student support | 69.5% | 63.0% | 53.4% |
| Students with mental health difficulties | Academics | 36.0% | 49.4% | 58.2% |
| | ALs | 19.7% | 13.7% | 15.0% |
| | Learning Tech | 17.5% | 20.5% | 46.3% |
| | Library | | 44.7% | 42.3% |
| | Student support | 74.4% | 79.6% | 67.2% |
| Students with mobility or manual dexterity issues | Academics | 58.2% | 67.5% | 63.3% |
| | ALs | 89.4% | 80.5% | 76.3% |
| | Learning Tech | 43.9% | 34.2% | 44.4% |
| | Library | | 55.3% | 50.0% |
| | Student support | 80.5% | 75.9% | 61.2% |
| Students with specific learning difficulties, such as dyslexia or dyspraxia | Academics | 52.5% | 61.2% | 63.3% |
| | ALs | 83.3% | 75.3% | 73.4% |
| | Learning Tech | 38.6% | 32.9% | 37.0% |
| | Library | | 44.7% | 61.5% |
| | Student support | 80.5% | 75.9% | 72.4% |
| Students with speech impairments | Academics | 52.5% | 55.7% | 52.6% |
| | ALs | 78.8% | 65.4% | 59.9% |
| | Learning Tech | 28.1% | 26.0% | 27.8% |
| | Library | | 39.5% | 46.2% |
| | Student support | 70.7% | 72.2% | 51.7% |
| Students with unseen disabilities and medical conditions | Academics | 46.0% | 56.1% | 53.6% |
| | ALs | 83.3% | 66.8% | 65.7% |
| | Learning Tech | 21.1% | 24.7% | 27.8% |
| | Library | | 23.7% | 26.9% |
| | Student support | 79.3% | 77.8% | 63.8% |
| Students with visual impairments | Academics | 59.8% | 62.4% | 61.2% |
| | ALs | 75.8% | 68.8% | 63.1% |
| | Learning Tech | 75.4% | 65.8% | 72.2% |
| | Library | | 60.5% | 76.9% |
| | Student support | 70.7% | 77.8% | 51.7% |

### 3.4. Support Contexts

Many of the respondents' perceptions of their support context stayed the same, showing no significant changes over time, despite the changing circumstances. The majority of staff in all groups consistently believe their colleagues are committed to accessibility (Q. 10), that the OU is committed to supporting disabled students (Q. 21), and that the OU actively works to encourage disabled students to succeed (Q. 22). However, there was a disappointing lack of improvement in staff knowing who to ask about accessibility issues (Q. 16) or being confident that they would get a useful answer in a reasonable timeframe (Q. 17), and a consistently low number of staff believe that OU documentation and guidance

are adequate for their purposes (Q. 15). Some of this may be related to pandemic-related disruption, as some staff groups showed non-significant rises in 2019, which then decreased in 2021. Data is shown in Table 8.

**Table 8.** Staff support context.

| Question | Staff Group | %Agree/Strongly Agree | | |
| --- | --- | --- | --- | --- |
| | | **2017** | **2019** | **2021** |
| 10. My colleagues are committed to accessibility (in their roles). | Academics | 81.2% | 86.3% | 85.2% |
| | ALs | 69.7% | 70.5% | 70.8% |
| | Learning Tech | 70.2% | 78.1% | 75.9% |
| | Library | | 92.1% | 92.3% |
| | Student support | 98.8% | 90.7% | 91.4% |
| 21. The OU is committed to supporting disabled students. | Academics | 88.9% | 92.2% | 89.3% |
| | ALs | 78.8% | 89.0% | 89.4% |
| | Learning Tech | 87.7% | 87.7% | 88.9% |
| | Library | | 89.5% | 96.2% |
| | Student support | 85.4% | 90.7% | 84.5% |
| 22. The OU actively works to encourage disabled students to succeed. | Academics | 79.3% | 83.9% | 82.7% |
| | ALs | 71.2% | 81.2% | 82.1% |
| | Learning Tech | 80.7% | 71.2% | 75.9% |
| | Library | | 84.2% | 96.2% |
| | Student support | 81.7% | 88.9% | 78.4% |
| 16. If I have questions about accessibility or inclusive practice, I know who I can ask. | Academics | 64.4% | 73.3% | 65.3% |
| | ALs | 63.6% | 68.2% | 69.7% |
| | Learning Tech | 66.7% | 53.4% | 64.8% |
| | Library | | 86.8% | 96.2% |
| | Student support | 82.9% | 81.5% | 74.1% |
| 17. If I have questions about accessibility or inclusive practice, I'm confident I'll get a useful answer in a reasonable timeframe. | Academics | 52.5% | 57.3% | 51.5% |
| | ALs | 39.4% | 57.5% | 56.6% |
| | Learning Tech | 49.1% | 46.6% | 61.1% |
| | Library | | 78.9% | 84.6% |
| | Student support | 67.1% | 68.5% | 57.8% |
| 15. The documentation and guidance provided by the OU in relation to accessibility is adequate for my purposes. | Academics | 45.2% | 46.3% | 44.9% |
| | ALs | 53.0% | 56.5% | 59.5% |
| | Learning Tech | 43.9% | 49.3% | 38.9% |
| | Library | | 63.2% | 57.7% |
| | Student support | 59.8% | 72.2% | 59.5% |

Staff perceptions of other aspects of their support contexts did change. For example, academics, Als, and learning technologists showed a significant increase in belief that accessibility training is appropriate for their roles (Q. 13) ($X^2$ (4, $N$ = 1844) = 10.894, $p$ = 0.028), and a significant increase in agreement from academics, ALs, learning technologists, and library staff that existing support structures for disabled students are fit for purpose (Q. 20) ($X^2$ (4, $N$ = 1844) = 12.888, $p$ = 0.012). However, responses from student support staff showed non-significant decreases in agreement to both questions 13 and 20, and they showed a significant decrease in agreement with Q. 23 that any barriers to accessibility they experienced were externally driven ($X^2$ (4, $N$ = 252) = 10.768, $p$ = 0.029). To the contrary, there was a significant increase across all staff groups in agreement with the statement that there are internal barriers to accessibility in the OU, i.e., driven by OU practices rather than external ones (Q. 24) ($X^2$ (4, $N$ = 1844) = 20.966, $p$ < 0.001). This increasing criticality of OU practices and recognition of internal barriers juxtaposes interestingly with the increased satisfaction with accessibility training; it may imply that the training is raising awareness of internal barriers or inaccessible practices. Results are shown in Table 9.

**Table 9.** Staff perceptions of barriers, systems, and training.

| Question | Staff Group | %Agree/Strongly Agree | | |
|---|---|---|---|---|
| | | 2017 | 2019 | 2021 |
| 13. The training available to me on accessibility is appropriate for my role. | Academics | 33.7% | 39.6% | 41.8% |
| | ALs | 43.9% | 44.2% | 55.1% |
| | Learning Tech | 33.3% | 35.6% | 40.7% |
| | Library | | 78.9% | 76.9% |
| | Student support | 62.2% | 63.0% | 48.3% |
| 20. The existing support structures within the OU for students with disabilities are fit for purpose. | Academics | 27.6% | 35.3% | 33.2% |
| | ALs | 43.9% | 39.0% | 49.6% |
| | Learning Tech | 21.1% | 26.0% | 33.3% |
| | Library | | 31.6% | 42.3% |
| | Student support | 47.6% | 46.3% | 37.9% |
| 23. Barriers to accessibility for students in my subject/specialist area are externally driven (e.g., accrediting bodies, technical constraints, etc). | Academics | 26.4% | 33.3% | 25.0% |
| | ALs | 24.2% | 26.0% | 28.1% |
| | Learning Tech | 22.8% | 17.8% | 18.5% |
| | Library | | 36.8% | 38.5% |
| | Student support | 26.8% | 35.2% | 19.0% |
| 24. There are internal barriers to accessibility in the OU (i.e., driven by OU practices, not external ones.) | Academics | 29.1% | 45.9% | 42.9% |
| | ALs | 22.7% | 37.3% | 29.6% |
| | Learning Tech | 29.8% | 43.8% | 44.4% |
| | Library | | 52.6% | 57.7% |
| | Student support | 47.6% | 42.6% | 56.9% |

## 4. Discussion

The survey results reveal a fascinating insight into the things that changed and the things that stayed the same throughout a turbulent 5-year period in an institution's journey towards greater equity and inclusion. In this section, we explore the changes, both positive and negative; the areas that did not change, with both positive and negative implications; and the differences and similarities between different staff groups. In doing so, we explore how these findings can inform whole-institution approaches to accessibility and inclusion.

The most significant changes between survey iterations can be summarised as increases in knowledge and critical awareness in relation to accessibility and inclusion. This was borne out through an increase in tested knowledge about disability by all staff, an increase in most staff groups' satisfaction with the training available to them, and an increase by all groups in critical awareness of barriers that exist in relation to inclusion, both externally in the sector and internally in the institution. The increases in knowledge are likely in response to a combination of internal interventions, such as training, support and comms, and external activity, such as an increasing discourse and strategic prioritisation around access, success, and inclusivity in the sector. It is interesting that perceptions of the institution did not become more positive as awareness increased, but rather became more critical over the years, as awareness improved. A key implication for the sector, therefore, may be that *the more awareness and knowledge staff have* around inclusion, *the more critical they become* of the institution's practices and processes. This juxtaposes interestingly against findings from Carballo et al. [25], Cook et al. [31], and Gelbar et al. [32], which found correlation between positive attitudes and low levels of practical knowledge relating to accessibility and inclusion in staff [24].

Changes over time relating to staff confidence are more subtle, as the survey results reveal a seesaw effect of a decrease in confidence in previously highly scoring groups and an increase in confidence in groups that previously scored lower. Some of the increases appear to be in response to targeted interventions; for example, the increases in confidence in supporting students with mental health difficulties align with internal training initiatives [18–21], and the increase in learning technologists' confidence likely relates to the aforementioned support initiative for learning technologists [23,24]. The decreases in confidence are harder to explain; it may be that training interventions focusing on some

groups of staff inadvertently excluded others, which combined with staff turnover may have resulted in drops in confidence. External factors may also play a part; it is likely that the drop in confidence shown by learning technologists between 2019 and 2021 may have related to the demands of the new accessibility legislation. It may also be a response to the increasing challenges faced by students throughout the period of the pandemic that may have exacerbated existing disabilities, and which staff with pastoral responsibilities (e.g., ALs and student support staff) will have experienced first-hand. Alternatively, studies have found a relationship between staff burnout and decreases in confidence [33], so it may be that increasing sector demands, and the emotional demands of the pandemic, played a role in this decrease. A key implication for the sector, therefore, is to be aware that *confidence levels can fluctuate*, and that *sustained, ongoing maintenance work is needed* to ensure staff confidence levels can be maintained in relation to inclusive practice.

Other crucial findings relate to the areas that did not change over time; namely staff positive attitudes and commitment to accessibility; and their less positive levels of confidence in their skills, and perceptions of their support context. Given the turbulent times the survey iterations spanned, it is commendable that attitudes and levels of commitment to accessibility remained high. This aligns with Lydon and Zanna's 'value affirmation' approach, in which people can maintain commitment in the face of adversity when that commitment is in line with their personal values [34]. However, the lack of improvement in staff support context (i.e., knowing who to ask about accessibility issues, being confident that they would get a useful answer in a reasonable timeframe or believing that OU documentation and guidance are adequate for their purposes), and the lack of improvement in areas where levels of knowledge were low are concerning. This may be an impact of the pandemic; as noted in the findings, staff showed non-significant increases in 2019, which then reduced in 2021, and studies are highlighting the detrimental effect of the pandemic on global equality issues [35]; it may also be indicative of a gap between policy and implementation (not uncommon in educational contexts [36]), and clearly needs to be an area of immediate focus for our institution. A key implication, therefore, is that *an institution culture that values positive attitudes and personal commitment to accessibility is essential in the journey towards inclusion*, and that this culture can *buffer potential institutional, sector and global challenges.*

The findings and implications from this study may be translated into four core recommendations for practitioners.

Recommendation 1: Institutions should place value on training and awareness raising, even if it leads to staff being increasingly critical of the institution. If it is effective, it should not lead to a decrease in individual commitment. However, it should be ongoing and sustained, supporting staff development across the board.

Recommendation 2: Institutions should prioritise (and allocate budget for) interventions and activity that promote inclusivity, as these do have potential to enhance practice. However, these interventions should be designed to be inclusive for different participants and should not be so targeted as to inadvertently exclude particular staff groups.

Recommendation 3: Institutions should promote a strong and holistic culture of commitment to inclusivity and accessibility, as staff individual personal commitment can make institutional inclusion-related activity more resilient to changes and challenges.

Recommendation 4: Institutions need to support their staff to understand who they can contact to resolve accessibility issues that go beyond their own knowledge and remit, and to have clear understanding of the institutional processes, roles, and responsibilities around accessibility.

This study has a number of limitations. As with many educational studies, it took place in a single institution, and results may not be generalisable to the broader sector (although the method could be adopted elsewhere.) Due to current events, the number of responses from tutors in 2017 was lower than in 2019 and 2021, and this may have affected the trends from this group. Additionally, this institution was large and complex; throughout the study, and particularly during the pandemic, institutional activity around inclusion was wide-ranging, and the authors may not be aware of the full range of interventions

and activity taking place. Furthermore, in the analysis of this data, we generalised and extrapolated from circumstances and staff groups, when the reality is likely to be much more nuanced and complex. Of course, the study is subject to the usual survey-related biases; respondent were self-selecting, leading to potential volunteer bias, and although questions were carefully framed, there is always the risk that respondents may not be entirely accurate in their answers (response bias.). And finally, disabled students were not involved, either in the design of the survey or in the study itself; it would be valuable to engage disabled student voices in future survey iterations.

## 5. Conclusions

This paper presented survey responses from staff surveys conducted in 2017, 2019, and 2021; it discussed changes over time and between different staff groups in their attitudes, knowledge, skills, and support contexts around accessibility and inclusion. In doing this, it identified key implications and recommendations for the sector that may support institutions in their journeys towards greater accessibility and inclusion.

It is evident from this study that our institution's journey to inclusion may be compared to a ship in stormy uncertain waters, buffeted by winds of change. The crew vary and fluctuate in their confidence, but as they gain experience they become increasingly aware of the risks of the sailing and how rough the sea can be. However, despite the challenges and the rigours of the journey, despite storms and insufficient hands on deck, the commitment to reach the destination, an equitable and inclusive learning environment, is unwavering.

**Author Contributions:** Conceptualization and methodology, K.L., V.K.P. and T.C.; software, K.L., V.K.P., T.C. and F.T.; formal analysis, K.L., V.K.P., T.C. and F.T.; writing—original draft preparation, K.L. an V.K.P.; writing—review and editing, K.L., V.K.P., T.C. and F.T. All authors have read and agreed to the published version of the manuscript.

**Funding:** This research received no external funding.

**Institutional Review Board Statement:** The study was conducted in accordance with the Declaration of Helsinki, and approved by the Human Research Ethics Committee of the Open University (protocol code HREC/2568, date of approval 12 May 2017).

**Informed Consent Statement:** Informed consent was obtained from all subjects involved in the study.

**Conflicts of Interest:** The authors declare no conflict of interest.

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
