# Peer review of "Inclusion in Uncertain Times: Changes in Practices, Perceptions, and Attitudes around Accessibility and Inclusive Practice in Higher Education"

_education, doi:10.3390/educsci12080571_

Round 1

Reviewer 1 Report

The article offers important data and analysis from three iterations of a biennial staff survey conducted in 2017, 2019, and 2021 at the British Open University to gauge various indicators of accessibility and inclusive practice before and during the Covid-19 pandemic. This exploration is attempted through periodic surveys administered to the staff members on agreed-upon and approved markers of leadership, resource, systems, and culture. The study explains how staff practices and perceptions change over time but certain characteristics remain constant. The examination also offers suggestions for institution-wide changes for accessibility and inclusive education.

Here are some further observations/recommendations:

1.     Literature Review: A little more context building and setting the stage from similar other research through literature review will help make richer comparisons.

2.     Methodology:  

·      Some of the return rates from the 2017 surveys are low. Some substantiation is warranted on this front.  

·      The research spans pre-Covid and during Covid circumstances. The difference in the result and impact of the pandemic may be highlighted as this will be great learning for global educational institutions. Similarly, the gap between policy and implementation can be exemplified further too.  

3.     Results:

·      Findings and correlates described well.

·      As a next step: Student voices, particularly from the disabled category would add value to the research and its holistic implications on institutions of higher education.

This is an interesting and relevant research of the Covid-19 period, where during a series of lockdowns almost all the educational institutions had to pivot to online learning quickly. The OU already had a well-established e-learning system that could be leveraged to implement accessibility and inclusive practices.

Overall, an essential contribution to understanding factors impacting accessibility and inclusive practices in higher education with corresponding policy implications.

Author Response

Thank you for your review! I really appreciate your insights, and I’m confident that the changes made in response to your suggestions have improved the article.

Here are a list of my responses to your points:

  1. Literature Review: A little more context building and setting the stage from similar other research through literature review will help make richer comparisons.

Thank you – I’ve added more literature on this in the introduction

  1. Methodology:  
  • Some of the return rates from the 2017 surveys are low. Some substantiation is warranted on this front.  

Thank you, you’re right. I’ve added a note on this in the limitations paragraph.

  • The research spans pre-Covid and during Covid circumstances. The difference in the result and impact of the pandemic may be highlighted as this will be great learning for global educational institutions. Similarly, the gap between policy and implementation can be exemplified further too.  

Thank you – I’ve added a reference to gaps between policy and implementation, and I’ve added to the text on the impact of the pandemic.

  1. Results:
  • Findings and correlates described well.
  • As a next step: Student voices, particularly from the disabled category would add value to the research and its holistic implications on institutions of higher education.

Yes, that’s a good point. I’ve added a note on that in the limitations, thank you.

Thank you again for your review!

Reviewer 2 Report

The theme is interesnting and very actual. Disscution is appropiate and the recommandation is vell done.  THe references must to be improuved. 

Author Response

Thank you for your review – as you suggested, I’ve added some additional literature and more references in the literature review.

Reviewer 3 Report

This paper examined changes in staff groups’ attitudes, knowledge, and skills in the context of accessibility and inclusion in 2017, 2019, and 2021 at a UK distance university. The authors report, among other items, significant increases in respondent knowledge and awareness of accessibility and inclusion. The authors credit this increase to internal interventions. Recommendations, based on findings, are provided by the authors.

Overall, the reviewer believes this paper is well written and includes institutional data and interpretations that demonstrate an institution’s progression for increased accessibility and inclusion. This paper provides specific value in demonstrating that all members of an institution play a key role in improving student accessibility and inclusion, a component often overlooked as support staff has been studied to a lesser extent than traditional faculty. The reviewer believes that the context and methods of the study are appropriate, as is the reporting of data. Suggestions related to grammar, formatting, and context are provided below.

·      Line 18 remove “and”

·      Line 153-157, align bullet points

·      Line 405, you say 3 recommendations and then provide 4.

·      The audience of this journal is international, spell out acronyms prior to use to improve clarity (e.g., HE, HEIs).

·      Tables – consider revising the first row of the tables/table headings for improved clarity (e.g., add question label, staff group label, move % agree/strongly agree above years). Consider adding “total or all” as an additional row for the staff group. This would provide a good visual to see how all cumulative responses varied from the years studied.

·      The context of the interventions is limited. Providing additional information about them could be helpful as this paper is presented as a case study. However, it does appear that some interventions have been described in more detail in previous literature and that literature is provided in references.

·      Reporting additional literature on support staff’s knowledge, attitudes, etc. on accessibility and inclusion would be helpful, as well as any studies that have examined interventions. There is a lack of connection between this study and previous literature examining similar phenomenon.

·      Consider addressing turnover of staff, within-group differences between years, etc. in the limitations section.

·      Consider adding additional statements/remarks, based upon the data, where this institution should focus intervention next to improve accessibility and inclusion.

Thank you for your contributions to this research space. This study was enjoyable to read and review.

Author Response

Thank you very much for your extremely helpful, thoughtful and kind review! I’ve made all the changes you suggest, and they have definitely enhanced the paper. I really appreciate you taking the time to do this.

Below are a list of responses to your points:

  • Line 18 remove “and”

Done, thank you

  • Line 153-157, align bullet points

Done, thank you

  • Line 405, you say 3 recommendations and then provide 4.

Apologies, there were originally three. I have changed this.

  • The audience of this journal is international, spell out acronyms prior to use to improve clarity (e.g., HE, HEIs).

Done, my apologies for not having done so before.

  • Tables – consider revising the first row of the tables/table headings for improved clarity (e.g., add question label, staff group label, move % agree/strongly agree above years). Consider adding “total or all” as an additional row for the staff group. This would provide a good visual to see how all cumulative responses varied from the years studied.

That’s a really helpful suggestion, thank you! I’ve made this change to the tables.

  • The context of the interventions is limited. Providing additional information about them could be helpful as this paper is presented as a case study. However, it does appear that some interventions have been described in more detail in previous literature and that literature is provided in references.

Thanks! I’ve added some more detail on the interventions and also signposted this section a bit more clearly.

  • Reporting additional literature on support staff’s knowledge, attitudes, etc. on accessibility and inclusion would be helpful, as well as any studies that have examined interventions. There is a lack of connection between this study and previous literature examining similar phenomenon.

Thank you – I’ve added more literature on this.

  • Consider addressing turnover of staff, within-group differences between years, etc. in the limitations section.

I can’t access institutional attrition data, unfortunately, but I have made a reference to staff attrition in the participants section. These were only samples of staff, so it’s likely that the samples will have returned different staff each time regardless of attrition.  

  • Consider adding additional statements/remarks, based upon the data, where this institution should focus intervention next to improve accessibility and inclusion.

Thank you – I’ve added that the next area of focus for the OU needs to be guidance and human support, as this appears to be the clearest gap between policy and implementation.   

Thank you again for your very encouraging review!